# Estimation of joint return periods of compound precipitationdischarge extremes for small catchments.

Ivan Vorobevskii, Rico Kronenberg, Christian Bernhofer

Faculty of Environmental Sciences, Department of Hydrosciences, Institute of Hydrology and Meteorology, Chair of Meteorology, Technische Universität Dresden, Tharandt, 01737, Germany

Correspondence to: Ivan Vorobevskii (ivan.vorobevskii@mailbox.tu-dresden.de)

#### Abstract

Compound hydro-meteorological extreme events represent a simultaneous occurrence of extremes or one extreme triggering another. These events do not necessarily require extremes in any of the components, but their combination could lead to an

- extreme. This study brings insights on compound extremes from the watershed scale perspective by applying a multivariate distribution approach to estimate return periods of compound precipitation-discharge extremes. Main research question concerned the degree of the agreement between extremes in terms of joint return periods (JRP). Additionally, impacts of spatial generalization for copula functions ('best-per-area') and outstanding extreme events were investigated and finally an attempt to correlate JRP with catchment characteristics was performed.
- The study was elaborated using data of small catchments from 5 to 675 km<sup>2</sup> (with median value of 103 km<sup>2</sup>) in Saxony (Germany) interpolated on a 1x1 km grid. A specially designed 'block-maxima' subset was defined to account for cases when the annual daily discharge and the annual daily precipitation maxima do not occur on the same day. 20 copulas were tested to find the one, which fits 'best-per-grid'. Afterwards, the most frequent copula ('best-per-area') was chosen to estimate JRP for compound extreme events. Additionally, a pool of the most frequent copulas for the study area was used to
- create an ensemble and calculate various quantiles of JRP. All chosen copulas have undergone two goodness-of-fit tests to account for the legitimacy of the approach, which result in very low percentage of rejection for both statistics. Overall, the approach shows a good agreement between precipitation-discharge extremes and a high potential for a probabilistic instead of a deterministic analysis of JRP. For the investigated catchments, compound precipitation-discharge extremes are highly correlated. Large uncertainty in the estimation of the JRP was revealed by comparing different copulas.
- This uncertainty increases with larger non-exceedance probability of the compound event. Few spatial patterns with either very low or high anomalies of JRP were identified. A 'best-per-area' distribution can be detected based on a frequency analysis of the 'best-per-grid' distributions. Nevertheless, in case of time-series shortage it could be more compelling and reliable to create an ensemble of the most frequent copulas since it allows already a probabilistic instead of a deterministic estimation of JRP. It is concluded that outstanding extreme events, which occurred in 2002 and 2013 have a significant
- effect on the compound extreme statistics. Finally, the catchment response seems to be largely independent of size and mean

elevation of the catchment, as no clear correlations between those and the corresponding JRP were found.

#### **1** Introduction

Compound extremes could be defined as an interdependence between two or more hazard drivers, which should not necessarily be extreme events individually, but trigger a significant extreme impact (Leonard et al. 2014; Mehran et al. 2017;

- Sadegh et al. 2018; Wahl et al. 2015). A phenomenon may occur as a result of one of the following situations (Field et al., 2012):
  - two or more simultaneous or successive extreme events (e.g., simultaneous extreme precipitation and storm surge (Moftakhari et al. 2015)),

•

- combinations of extreme events with underlying conditions that amplify the impact (e.g., droughts and heatwaves (Mazdiyasni and AghaKouchak, 2015)),
- combinations of events that are not themselves extreme but collectively lead to an extreme event or impact (e.g., a moderate coastal flood occurring during above average tide (Moftakhari et al. 2015).

Recently compound extremes started gaining more and more attention by researchers, still the number of publications comparing to univariate extreme studies is very low (less than 5 % according to database from (Web of Science, 2019)). Observations and simulations have been used to explore the relationship between multiple variables/components of

- compound extremes. Due to limited observations of extremes (or rare events), the statistical inference of compound extremes and the extrapolation beyond observations are the most commonly used tools (Hao et al., 2018) and is a baseline methodology of the presented research. These methods include empirical approaches, multivariate distributions (Hao and Singh, 2015a), indicator approaches (Kew et al., 2013a), quantile regressions (Hirschi et al., 2010), and Markov Chain
- models (SedImeier et al., 2016). Several researchers indicate a potential increase of compound event occurrence and magnitude due to climate change (Hao and Singh, 2015b; Little et al., 2015; Serinaldi, 2016; Zscheischler and Seneviratne, 2017) thus the assessment of climate change impact on compound extremes is of particular importance for adaptation measures due to their tendency to have a larger impact than an individual extreme (Field et al., 2012). Another approach of compound hydro-meteorological events analysis is conceptual and physical based simulations, though only a few researchers
- have studied this field recently (Hurk et al., 2015; Kew et al., 2013b; Pasquier et al., 2018; Zscheischler et al., 2018). The majority of modern statistical assessments of compound events are referring to either meteorological (i.e. compound extreme temperature, heat wave and heavy precipitation (Tencer et al., 2014)) or hydrological (i.e. flood peak, duration and volume (Ganguli and Reddy, 2013)) extremes or limited to big catchments (i.e. compound precipitation and discharge deficits for Rhine basin (Beersma and Buishand, 2004) or large areas using coarse grid datasets in general (i.e. flood hazard
- combined with precipitation and temperature extremes in Norway (Benestad and Haugen, 2007)). Currently there is a lack in studies which consider compound hydro-meteorological extremes by taking a closer look on what is happening at a small-catchment scale. However, few researchers have already pointed out on the specifics of small and big watersheds regarding

75

extreme events. For the precipitation-discharge relationship one of the common and well-known feature of small catchments is that they could expose more frequent and much more abrupt with higher relative magnitude extremes in comparison to

- bigger ones (Dadson et al., 2017; Marchi et al., 2010; Westra et al., 2014) which have higher retention and regulation capacity. Moreover, global assessment of flood and storm extremes (Wasko and Sharma, 2017) pointed out that effects of precipitation losses on runoff generation due to climate change are reduced in smaller catchments. Heavy precipitation intensity may remain the same and the peak streamflow is more likely to be influenced by precipitation characteristics rather than by the catchment wetness pre-conditions, hence a higher correspondence between precipitation and discharge extremes
- is expected for smaller catchments. Still, due to large heterogeneity in the factors rainfall-runoff process (i.e. relief, land use, soils) between catchments, detection and research of hydro-meteorological extremes on a spatial scale using statistical methods becomes more difficult due to the necessity of application of areal interpolation methods (thus imminent smoothing of local patterns and features) to obtain data.

The main objectives of this study are:

- The application of multivariate distribution approach to study return periods of compound precipitation-discharge extremes in small catchments.
  - Conclusions on the applicability and uncertainty of the method and specifics of compound extremes in Saxony (Germany).

Research questions of the paper are stated as following:

- How do annual maximum precipitation and discharge intercorrelate in small catchments in terms of joint return periods?
  - What are the impacts of spatial generalization for copula functions and outstanding extreme events on the estimation of joint return periods?
  - Are there any patterns in the relation between catchment characteristics and compound extremes?
- The paper is divided in four parts. The introduction to the general topic is followed by the description of the study area and the procedure to obtain gridded datasets for precipitation and discharge (section 2.1), then the main methodology part of the copula approach is presented (section 2.2). Results contain features of best-fitting copulas (section 3.1), joint return periods for design events and influence of outstanding extremes (section 3.2), uncertainty of joint return period estimation using a copula ensemble (section 3.3) and correlation with catchment characteristics (section 3.4). Finally, a conclusion and an
- outlook are drawn in section 4.

#### 2 Methods and data

#### 2.1 Precipitation and specific discharge datasets

Gridded daily precipitation data set (P) was obtained from RaKliDa (Kronenberg and Bernhofer, 2015) which is a freely

- available and regularly updated (2 times per year) climatological dataset for several regions in Germany (for the period 1961-2018) developed at the Chair of Meteorology at Technical University of Dresden. Initial meteorological information 95 comes from stations operated by the German Meteorological Service and the Czech Hydrological Meteorological Institute and then corrections of wind dominated errors (Richter, 1995) are applied. Afterwards, point data is interpolated with indicator Kriging (Wackernagel, 2003) on a 1x1 km grid. The approach is stated to reflect the orographic influence of lee and wind ward effects as well as to account properly for the convective and small-scale precipitation events. Approximately 100 150 stations for the chosen part of Saxony are used.
- To obtain a specific discharge data set (SD) initial data from 87 relative small catchments (Figure 1) with drainage areas from 5 to 675 km<sup>2</sup> (mean value 141 km<sup>2</sup>, median value 103 km<sup>2</sup>) was retrieved from the Saxon official online database (Sächsisches Landesamt für Umwelt, Landwirtschaft und Geologie, 2019a). This number corresponds to network density of 1 gauge per approximately 150 km<sup>2</sup>. Estimations of mean values for the whole time series (annual mean and annual maxima
- of specific discharge) (Figure 2) show a big range, especially for maximum flow (from 20 to 248 ls<sup>-1</sup>km<sup>-2</sup>), however with a clear tendency of increase of the both discharge statistics towards mountainous regions. Additional a pre-analysis of the data availability for the chosen gauges showed that after 1960<sup>th</sup> the percentage of missing data drops shortly from 80 to 40 % and the lowest values of 5 % were founded in 2000<sup>th</sup>. So it is acceptable to use 1961 as a common starting point for both datasets. There are few commonly used geo-statistical methods in hydro-meteorology for the spatial interpolation of precipitation and
- discharge (Basistha et al., 2008; Buytaert et al., 2006; Hisdal and Tveito, 1992; Hong et al., 2017; Paiva et al., 2015; Parajka 110 et al., 2015; Yoon et al., 2013; Zhang et al., 2014): inverse distance weighting (IDW), ordinary kriging (OKR), universal kriging (UKR). Since there is no general agreement on whether one of them in superior and no publications were found on the daily runoff interpolation which could be applicable in the study case, it was decided to test all of common approaches with a 'leave-one-out' cross-validation (Figure 3). The following parameters were used for IDW: minimum number of
- gauges 3, maximum radius 40 km, inverse distance power 2; for OKR: automatic variogramm choice (between 115 exponential, spherical, Gaussian, Matern models), for UKR: mean catchment elevation as a covariate and an automatic variogramm choice. Additionally, a mean out of all three methods was calculated (ensemble). Results were analyzed with respect to conventional performance indices used in hydrology: mean absolute error (MAE), root mean square error (RMSE) (Legates and McCabe Jr., 1999), Nash-Sutcliffe efficiency (NSE) (Nash and Sutcliffe, 1970) and Kling-Gupta-efficiency
- (KGE) (Gupta et al., 2009). All tested methods showed minor differences in performance both for the whole time-series (mean values: RMSE around 7 ls<sup>-1</sup>km<sup>-2</sup>, MAE around 3 ls<sup>-1</sup>km<sup>-2</sup>, NSE and KGE around 0.7) and annual extremes (mean values: RMSE around 40 ls<sup>-1</sup>km<sup>-2</sup>, MAE around 30 ls<sup>-1</sup>km<sup>-2</sup>, NSE and KGE around 0.7) with a slight advantage for the ensemble and IDW. While NSE and KGE values state a relative good efficiency of methods (according to accepted benchmarks (Knoben et al., 2019; Ritter and Muñoz-Carpena, 2013)), the comparison of the MAE and RMSE outcomes with
- mean and maximum specific discharge for each of the tested catchment (Figure 2) reveal a large relative error. On average 125 for all validated gauges, MAE and RMSE for the whole time-series amount 37 % and 68 % of mean specific discharge values respectively, while for annual maxima errors they yield to lower values -32 % and 48 % of mean annual specific

discharge values respectively, implicitly indicating a potentially better predictability of extremes, which is the main intention of the whole interpolation procedure. However, it does not imply a bad performance since we compare the mean error with

- the mean discharge. Therefore, based on the cross-validation results, significantly smaller computational time and also no significant correlation to external variables (altitude) IDW method was chosen for 1x1 km grid interpolation. Evident outlier catchments (5 in total) were manually checked and three of them were excluded (as evident signs of heavily anthropogenic changes were found: reservoirs, mining, high urbanization, deforestation) from the final dataset, yet others are representing either very small or significantly remote catchments and were decided not to be removed as no clear evidence of notable
- human influence was discovered.

For the presented study an annual conventional 'block maxima' method (Gumbel, 1958) was chosen to subset annual maximum precipitation and specific discharge (58 pairs per grid cell as maximum). Since the application of 'peak-over-threshold' method (Leadbetter, 1991) on a gridded dataset will lead to ambiguous results due to the spatial variability of the threshold value as it will be unique for each grid cell. The first main objective of this study implies to find compound return

- periods of exceedance a threshold on the same day. Therefore, a subset created in this manner will not be enough, since in many cases the annual daily maxima of specific discharge and precipitation will not occur on the same day (sometimes not even in the same month). For this reason, a straightforward analysis of this subset will only give a probability of non-exceedance within the same year, but not on the same day. Thus, it is reasonable to create two datasets:
  - daily annual maximum precipitation and a maximum out of a 2-day specific discharge (occurring on the same day as precipitation and the day after) and second; consequently,
  - daily annual maximum specific discharge and the maximum of a 2-day precipitation (occurring on the same day as specific discharge and the day before).

For the future simplification these datasets will be named as Subset 1 (Pmax+2daySD) and Subset 2 (SDmax+2dayP). The 2day flexibility for the paired-variables is kept to account for a few reasons. The first possible reason is a delay caused by the rainfall-runoff generation processes, the second one refers to the morning floods caused by late evening rainfall and the third

one concerns differences in sub-daily recordings (thus calculations of mean daily values) for meteorological stations and hydrological gauges. Therefore, applying these two datasets one can estimate compound return periods of a certain event occurring on the same day by taking a mean value out of the two estimates from both datasets.

#### 2.2 Multivariate distribution and estimation of return periods

- The multivariate distribution approach via copulas was used to access interdependency between precipitation and specific discharge since it is the only statistical method which defines correlation as non-constant value and allows to have a look beyond observations' range by fitting a theoretical model. For two random variables X and Y with defined marginal distributions  $M_X$  and  $M_Y$ , respectively, the joint bivariate cumulative distribution function F(x, y) (meaning non-exceedance probability P of x or y thresholds for variables X and Y respectively) can be expressed with a copula C with some
- parameters  $\theta$  (Nelsen, 2006; Sklar, 1959):

180

190

## $F(x, y) = P(X \le x, Y \le y) = C(M_X(x), M_Y(y), \theta)$ (1)

There are several general copula families existing (Hao et al., 2018), which are commonly used in environmental sciences for the construction of multivariate distributions, there are: elliptical (i.e. Gaussian, Student-t), Archimedean (i.e. Frank, Clayton, Gumbel, independence, Joe, BB, Tawn), and extreme-value copula (Gumbel-Hougaard, Galambos, extremal-t, Hüsler-Reiss). All of them show different properties in the modelling of the dependency structure (i.e. symmetric/asymmetric, lower/upper tail dependency) thus allow to find a theoretical function close to the shape of the cloud of observed points. Both empirical and theoretical (one-, two- and more-parameter) copulas have been successfully applied in hydro-meteorological studies (Fan et al., 2017; Favre et al., 2004; Salvadori and Michele, 2007; Zhang et al., 2012).

The fitting of a chosen copula is based on a transformation of the initial dataset of variables to so-called pseudo-observations  $U_X$ , which for some random variable X is defined as a simple ranked normalization of each realization:

$$U_X(x_i) = r(x_i)/(n+1)$$
 (2)

where r denotes the rank of  $x_i$  and n is the length of X. Afterwards the copula parameters are estimated with the canonical maximum likelihood method. Finally, this procedure is repeated for multiply copula types. Subsequently AIC (Akaike, 1974) and BIC (Schwarz, 1978) values are calculated and compared and based on the result one can decide on a 'best-fit'

type for the specific dataset. For the fitting process the R-packages 'copula' (Yan, 2007), and 'vinecopula' (Schepsmeier et al., 2018) were used.

There are several tests available to assess the goodness-of-fit of a specific fitted copula to the data. One of the most widely used and recommended (Genest et al., 2009) as a blanket procedure for all copulas types is based on a the difference between the chosen theoretical  $C_{\theta n}$  and the empirical copula  $C_n$  using pseudo-observations U with n pairs and d dimensions. Using Cramer-von Mises statistics  $S_n$  value can be calculated as follows:

$$S_n = \int_{[0,1]^d} (\sqrt{n} (C_n(U) - C_{\theta n}(U)))^2 \, dC_n(U)$$
 (3)

With H<sub>0</sub> denoting to non-rejection of tested theoretical copula, the p-value for the obtained test statistic can be calculated from the distribution of bootstrapped  $S_n$  (with a slight change of the copula's parameters: 200 replicates were chosen as an appropriate number for the study) using a specially adopted Monte-Carlo methods (as limiting distribution for  $S_n$  depends on

the copula family). Modification of the abovementioned test is called  $S_n(b)$  and is stated as more powerful. It has a similar calculation procedure as  $S_n$  in general, but is based on the Rosenblatt's transformation of the initial data (Genest et al., 2009).

The return period of a given event is usually defined as the average time elapsing between two successive realizations of the event. For univariate cases it can be assumed a random variable *X* has a distribution function M(X). Its non-exceedance return period URP(x) is than defined as:

$$URP(x) = \frac{1}{1 - M(x)} \qquad (4)$$

In the bivariate case (or multivariate in general) two types of recurrence intervals are usually considered (Salvadori et al.,

2007): based on joint non-exceedance probabilities 'OR' (one/more variables) and 'AND' (both/all variables). In the context of the presented study the latter one is used (Serinaldi, 2015). For random variables X and Y the joint return period  $JRP_{AND}(x, y)$  can be calculated as follows: 195

$$JRP_{AND}(x,y) = \frac{1}{1 - M_X(x) - M_Y(y) + C(U_X(x), U_Y(y))}$$
(5)

where  $M_x(x)$  and  $M_y(x)$  are the marginal distributions of two variables and  $C(U_x(x), U_y(y))$  is the copula function.

Figure 4 illustrates the possible range of joint return periods in comparison to univariate return periods. Joint return periods depend on the Pearson correlation coefficient (R) between random variables (X1 and X2) by using an empirical copula. The 200 closer the agreement between the variables is, the closer is the JRP to URP: indeed, if one considers events with a 10 year URP (identical for both X1 and X2) JRP is estimated up to infinity for negative correlation (for R=-1), 100 years for complete independency (for R=0) and up to 10 years for positive correlation (for R=1). Despite the fact, that for various theoretical copulas the URP-JRP relationship will not be identical to the presented one (except for R values of -1, 0, 1), it can be clearly stated, that for a positive correlation and a bivariate case the JRP of a certain design event (here it means build-up using variables with identical URP) should be always in the [URP:URP<sup>2</sup>] interval.

#### **3 Results and discussion**

#### 3.1 Choice of the best fitted distributions

There are around 30 copula types available for fitting in the packages used. However, under a natural conditions it is reasonable to assume a positive correlation between precipitation and discharge maximum. Hence, one can narrow the list to the following types which are described by increasing functions: BB (1, 7, 8), Clayton, Frank, Gaussian, Gumbel, 210 Independence, Joe, Student-t, Tawn (1, 2) and their survival replica, which yields 20 copula types in total.

The presence of spatial heterogeneity of precipitation and discharge data obviously leads to heterogeneity in the best-per-grid copula findings. However, to apply the abovementioned approach and get reliable and consistent results (i.e. comparable return periods between grid cells) one copula type per dataset has to be selected for the whole territory in the end. The easiest

- way to reach the goal is to select the most frequent best-per-grid type. As it is depicted in Figure 5 both Subset 1 215 (Pmax+2daySD) and Subset 2 (SDmax+2dayP), as well as the reduced ones, clearly show dominant copula types. The subset with annual discharge as a lead variable (Subset 2) shows the higher agreement with survival Clayton and Joe copulas with a coverage of approximately 75 % of the territory. In fact, it can be concluded implicitly that the agreement is even higher, since both functions have a very similar shape with a clear upper tail dependency. On the other hand, the subset with
- precipitation as a lead variable (Subset 1) exhibits, in general, three types of dependency (upper, lower tails and 'normal' shape) thus results are not so consistent; maximum values of obtained frequencies are much lower (only approximately 40 % of the territory is covered by the two best copulas - Gaussian and Gumbel) and in contrast to Subset 2, extremes from Subset 1 has different dependency structure ('normal' shape and upper tail respectively). Additionally, it is found that the exclusion

of the outstanding flood events of 2002 and 2013 does not lead to a change in the best-per-area copula choice (Joe and 225 Gaussian copulas remain the most frequent for the two subsets respectively). Yet a considerable redistribution of frequencies happens. For the Subset 2 (SDmax+2dayP) a decrease in the upper tail dependency is observed, which is leading to a sharp decrease in frequencies for Joe and survival Clayton copulas. Features of the data such as normality (Gaussian copula) or asymmetry (Tawn 1 copula) are starting to play bigger role in the best copula selection procedure. A quite similar situation, yet less expressed, happens to the Subset 1 (Pmax+2daySD): precipitation and discharge extremes lose their upper tail 230 dependency (i.e. reduction of frequencies for Gumbel, survival Clayton, Joe copulas) and become more normal- and uniform-shaped (i.e. increase of frequency for Gaussian and Frank copulas respectively).

#### 3.2 'Best-per-area copula' approach and influence of extreme events

By applying Gaussian and Joe copulas to both Subsets 1 and 2 respectively and calculating the mean JRP of design events one can conclude on the agreement between precipitation and discharge extremes on a daily scale (Figure 6). It was found, that for all studied design events the maximum possible range of [URP:URP<sup>2</sup>] for JRP is almost fully covered. Yet, median values for the whole area are closer to the lower boundary: 24 years for 10-10-year event, 172 years for 50-50-year event and 403 years for 100-100-year event. Thus, it can be stated that in general a good correlation between both extremes is presented within the study area. Despite a fuzzy distributed outcome, two hotspots are consistent for all tested design events: the upper part of Weiße Elster basin (south-western part) with high values (i.e. 1000-6000 years JRP for 100-100-year event)

and the upper parts on Elbe tributaries (eastern part of Ore mountains) with values very close to URP (150-200 years JRP for 100-100-year event). Additionally, it can be highlighted, that the spatial heterogeneity of JRP is growing with increasing URP.

The removal of the outstanding extremes of 2002 and 2013 from the gridded datasets leads to a notable effect on the estimation of JRP. The median values of JRP increase by 10-22 % which could implicitly indicate the presence of a 'fake' correlation effect (for the cloud of points with weaker relationship several outliers could artificially and significantly raise

- correlation effect (for the cloud of points with weaker relationship several outliers could artificially and significantly raise the correlation). Moreover, the spatial patterns have changed: the hotspot with low JRP in the eastern Ore mountains gets less visible, while another hotspot in the Zschopau catchment (middle part) with JRP above average arises. Few studies exist on the statistical assessment and comparison of 2002 and 2013 extreme floods in Europe and in Germany in particular (Blöschl et al., 2013; Merz, Bruno et al., 2014; Schröter et al., 2015; Thieken et al., 2016; Zink et al., 2016).
- however the statistical analysis of precipitation and discharge was done separately and moreover gauge-based for a relatively big catchments, whereas the estimation of joint return periods on a spatial scale for small catchments is missing. Bivariate and marginal probabilities have to be estimated to find JRP of a particular (not design) event, therefore appropriate marginal distributions have to be found for each of the variables to construct a multivariate distribution. Using the same procedure as for the copula (maximum areal frequency of best-per-grid distribution according to AIC/BIC) several functions (normal, log-
- normal, logistic, log-logistic, Gumbel, gamma, Generalized Extreme Value, exponential, Weibull and Pearson-3) were tested. GEV (for annual maximum precipitation) and Pearson-3 (for annual maximum specific discharge, 2-day-precipitation)

maximum and 2-day-specific discharge maximum) were chosen as best ones. The results presented in Figure 7 indicate that the event in August 2002 was much stronger in terms of both heavy rain and flood which affected the central part of Saxony. The URP of the daily precipitation maxima were much higher (500-1000 years on average with maximum up to 17000 years) than the URP of the specific discharge (200-300 years on average with maximum up to 650 years) which lead to the large values of JRP up to 75000 years (15000 on average). The situation in June 2013 was however different. It was weaker and affected the western part of study area. Daily precipitation maxima barely reach 10 year URP, yet since previous rain events and relatively small flood events yielded to high spatial extension of fully saturated soils (Zink et al., 2016) quite an outstanding flood (200-500 years on average and maximum up to 1050 years) was generated with JRP of 1000-2000 years on average. This suggests under the same precipitation conditions higher pre-event soil moisture leads to higher URP of discharge and thus to higher JRP.

The last remark on Fig 6 and 7 will concern the white spot (approximately 50 cells) in the Spree catchment (eastern part) which finally was decided to leave in the figures to illustrate a numerical problem of the fitting process (i.e. it results from a failure to estimate Joe copula parameters due to zero Kendall Tau correlation). One of the most probable reasons for this

incomplete fitting is that the catchment (with a mass center is right in the middle of the spot) used for interpolation represents an outlier and the reasons in this case are unclear. Nevertheless, these findings could be treated as a third round on outlier check for the hydrological dataset.

#### 3.3 'Copula ensemble' approach

The selection process of the copula in section 3.2 is based on the maximal frequency of best-per-grid copula type. However, it may happen that this approach does not capture the real dependency structure between precipitation and discharge extremes for the territory. At first there is a strong possibility of very close AIC/BIC values estimated for the 'first-bestchoice' and other tested copulas for each grid. The second reason is that by sticking up with one copula for the study area ('best-per-area') in the end could lead to the smoothing of the spatial variability of dependency type (hence copula type) and loss of spatial patterns. Both effects become even more important when estimating probabilities of rare events (upper tail

dependencies).

One of the possible solutions to account for the uncertainty of estimated return periods is based on a copula ensemble. This means that instead of using only the best combination of copulas (Gaussian and Joe for Subset 1 and 2 respectively), one creates a pool of different copula types with high 'best-per-grid' occurrence and finally analyzes all combinations. In our case 5% of the spatial coverage (Figure 5) was chosen as a threshold to select Gaussian, Gumbel, Joe, Survival Clayton,

Survival Gumbel and Tawn 1 copulas for the Subset 1; Gumbel, Joe, Survival Clayton and Tawn 1 copulas were chosen for the Subset 2. This leads to 24 possible cross-combinations, and thus to an ensemble of 24 values of JRP. To account for the legitimacy of the approach all chosen copulas have undergone two goodness-of-fit tests. The percentages of territory with rejection of  $H_0$  (p<0.05) and a 'good' fit (p>0.90) for each of the chosen copulas and both tests are presented in Table 1. The results are based on 200 bootstrap replications. In general, the results show very low number of rejection

- values for  $S_n$  (up to 8.3 %) and  $S_n(b)$  (up to 1.7 %) statistics. This indicates that all tested copulas can be applied for the whole territory. On the other side, the analysis of the number of grids with high p-values (getting close to spatial acceptance of  $H_0$  (although that will mean p-values strictly equal to 1) did not show high values as well (up to 17 %), despite the fact that the in Figure 5 depicted Joe and Survival Clayton as clear dominant copulas for Subset 2. Yet, obtained results should be treated with caution, since 58 years (as a maximum) of data available could be not enough for a proper identification of a 'true' dependency structure (Genest et al., 2009) and therefore weakening the performance of goodness-of-fit tests'.
- The created ensemble allows to explore a possible range of JRP via low and high empirical quantiles and median values for design events. The analysis of the mean JRP of the study area for the 20th and 80th quantiles reveal a large possible range of JRP (in comparison to median JRP) which is increasing with URP: from 33 % for 10-10-year up to 97 % for 100-100-year event (Figure 8). Additionally, it is found that for the majority of grid cells and all design events a left-side skewness of the
- ensemble distribution occurs towards lower percentiles (i.e. median values are closer to the 20<sup>th</sup> rather than to 80<sup>th</sup> quantiles).
   Finally, a higher variance of JRP towards higher quantiles and higher URP was discovered.
   Several features could be uncovered by the comparison of the results gained with best-per-area copula (Section 3.2) and the ensemble approach. While Joe-Gaussian copula combination gives JRP close to the ensemble median values for a 10-10-year design event, it becomes almost identical with the ensemble 80<sup>th</sup> quantile for a 100-100-year event. This fact indicates
- that the ensemble approach assigns higher non-exceedance joint probabilities for median values for the same events and smooths the tail dependency by mixing various copula families. Furthermore, spatial patterns can be found, which are in general visible in both cases (south-western part with high and eastern part of Ore mountains with low JRP values).

#### 3.4 Joint return period and catchment characteristics

The intersection of the obtained gridded JRP for 100-100 year event with the shape of drainage areas for small catchments 310 (Sächsisches Landesamt für Umwelt, Landwirtschaft und Geologie, 2019b) allows the study of how basic catchment characteristics (mean elevation and area) possibly correlate with JRP (Figure 9a). The highest range of JRP denotes to the smallest catchments, while for areas larger than 10 km<sup>2</sup> the variation drops sharply. In could be seen, that the lower boundary of JRP is generally increasing with the catchment size. Two clusters of catchments can be distinguished by the application of elevation as a covariate: the upper part of Weiße Elster basin (south-western part) with JRP 1000-5000 years and lower parts

in the basins of Elbe, Großer Röder, Freiberger and Zwickauer Mulde with JRP 300-600 years. By summarizing the scatter plot to a histogram (Figure 9b) and overlapping it with JRP calculated without 2002 and 2013 it becomes apparent that the exclusion of outstanding extremes leads to a shift towards higher values and larger variance of JRP for the catchments.

#### 4 Summary and conclusions

The presented study investigates compound events by inco