# Peer review of "Estimation of joint return periods of compound precipitationdischarge extremes for small catchments."

_Hydrology and Earth System Sciences, 2020_

## Referee Comment (RC1) · Anonymous Referee #1 · 24 Feb 2020

**Review for manuscript "Estimation of joint return periods of compound precipitation-discharge extremes for small catchments"**

**Authors:** Ivan Vorobevskii, Rico Kronenberg, and Christian Bernhofer
**Journal:** Hydrology and Earth System Sciences

**Summary**

The authors compute joint return periods for a gridded dataset in Saxony, Germany derived from streamflow observations in 87 catchments. To do so, they use two precipitation-discharge datasets. The first dataset is generated by selecting precipitation conditioned on annual maxima discharge events and the second dataset by selecting discharge events conditioned on annual maxima precipitation events. They fit a large number of copulas to each of these datasets and use a subset of copulas for the estimation of event magnitudes corresponding to return periods of 10-, 50-, and 100-years. In addition, they determine the joint return periods of severe flood events observed in 2002 and 2013.

**General comments**

The study by Vorobevskii et al. addresses the question of how frequent joint flood (Q)-extreme precipitation (P) events are. While I see the relevance of determining recurrence frequencies for each of these extremes and for looking at joint probabilities of e.g. fluvial and coastal floods, I don't see the practical relevance of looking at the variable pair P and Q jointly in a 'compound' event framework. To me the study is lacking a convincing motivation and is currently neither contributing to an improvement of modeling strategies of joint extremes nor to process understanding. As it is presented now, this study seems to be an exercise of fitting copulas to a pair of variables without a clear motivation and without providing meaningful conclusions.

In addition, I see some severe issues in the methodology which may lead to conclusions imposed by certain model choices. First, the joint probabilities are computed using a gridded dataset derived from station observations using inverse distance weighting which might introduce some spurious relationships between Q and P. I do not understand what the value of using a gridded instead of a catchment dataset is here and would highly recommend to change to a catchment instead of a gridded dataset. Second, the two datasets (Q conditional on P and vice versa) are chosen in a conditional fashion while joint probabilities are computed. I would recommend to use a peak-over-threshold approach allowing for the selection of events where both variables exceed a threshold jointly. This would be consistent with the computation of joint probabilities and allow for a seasonal analysis which may lead to novel insights as compared to an annual analysis. Third, the nature of the dependence structure between P and Q, i.e. its intensity and form, should be investigated which would contribute to an understanding of how these two variables are related and allow for an informed choice of suitable copula functions. In addition to methodological improvements, the manuscript would also profit from some restructuring to improve the reading flow, language editing, and using appropriate color schemes in figures. For all these reasons, I can not recommend this manuscript for publication in HESS. Please find below a few more detailed comments and suggestions of how to improve the manuscript.

**Specific comments**

1. What is the practical motivation for studying the joint probability of Q-P events? For whom is this of interest?

2. What are the research questions of this study eventually leading to novel insights? The relationship between P and Q has been studied intensively from a process perspective and the methods used for bivariate frequency analysis used in this study are not novel either.

3. I would structure the introduction in the following way: compound events in general and why they are important, Q-P events and why they are important, gap, research questions, strategy of addressing research questions.

4. The manuscript would profit from language editing and mprovement of the reading flow.

5. I do not think that the size of the catchment is the only factor relevant for determining the Q-P relationship (l. 70). I think that it mainly depends on the importance of other potential flood triggers such as snowmelt and soil moisture availability.

6. The Methods section should start with an introduction to the study area.

7. The nature of the precipitation dataset remains unclear to me. The authors write that they downloaded a gridded precipitation dataset (l.93) but at the same time they state that they interpolated point data (l.97). Was precipitation data also interpolated using IDW?

8. The use of a gridded instead of a catchment dataset may introduce spurious correlations between P and Q. I would therefore use a catchment dataset (areal mean precipitation and streamflow at the catchment outlet) instead of a gridded dataset.

9. Is it correct that the dataset comprises years with missing data of up to 80% (l. 107)? If yes, I would reconsider to limit the time period to a sub period for which reliable data are available or to remove the catchments from the analysis for which very limited data are available.

10. In Figure 3, I would show relative errors instead of absolute errors.

11. The nature of dependence between Q and P should be investigated. What is its form and intensity? I would specifically look at Kendall's tau rank correlation coefficient [*Kendall*, 1937] to characterize the general dependence structure and on upper and lower dependence coefficients to understand whether dependence is symmetric/asymmetric and whether there is upper or lower tail dependence [*Poulin et al.*, 2007]. A better understanding of the dependence characteristics may guide the choice of suitable copulas in addition to the use of goodness-of-fit tests which are often not very reliable in the presence of small sample sizes [*Serinaldi et al.*, 2015]. The Clayton copula could for example be excluded because it represents lower tail dependence, which is hardly the type of dependence observed for Q-P pairs.

12. Events are chosen in a conditional fashion while joint instead of conditional probabilities are computed. From a process and impact perspective conditioning Q on P seems to be much more relevant.

13. I do not exactly understand how the copula functions were chosen. Was a goodness-of-fit test performed (as suggested in l. 177-189) in addition to the computation of the AIC and BIC criteria (l. 173-175)? The order should be adjusted. First, a gof test is performed in order to exclude rejected copulas, then, the copula with the lowest AIC or BIC values is chosen among the non-rejected copulas. Were AIC and BIC performance averaged?

14. I do not understand how the marginal distributions were chosen, fitted. Later in the results section, some fitting experiment is described. Was this applied for all the analyses presented or were empirical distributions used in some cases?

15. Joint return period definitions are not the only way of defining bivariate return periods (l. 191). Conditional or approaches based on the Kendall's distribution can be used instead (see e.g. [*Brunner et al.*, 2016; *Gräler et al.*, 2016]).

16. It is unclear how the results shown in Figure 4 were produced (l. 196-206). Was a simulation experiment performed or are these results based on observations? If a simulation experiment was performed, I would remove the part with negative correlations since P and Q are usually positively related.

17. It is not surprising that the dependence structures derived for the two datasets (Q conditioned on P and vice versa) are different. High Q usually requires some high P input. However, high P may not necessarily lead to high Q.

18. I think that the exercise of excluding important flood events from the computation of joint probabilities is meaningless. What is the idea behind this experiment? Of course tail dependence will decrease if severe events are excluded because the tail is cut off. I would remove this part of the analysis altogether.

19. I think that it is problematic to compute mean joint return periods derived from the two subsets (l. 233). The two datasets answer two different questions: what is the probability of joint exceedance given that Q is high and vice versa.

20. The reason for the failure of fitting a Joe copula to a dataset with zero correlation (l. 267-271) is not the failure of the Joe copula but the choice of an unsuitable copula. These datasets are apparently independent. In this case: F(x,y)=F(x)*F(y). No copula is required here.

21. I am not very much in favor of the ensemble approach presented here (Section 3.3). Instead of using as many copulas as possible I would rather try to constrain the number of suitable copulas by trying to understand the type of dependence that has to be represented by the copula (see also one of my previous comments).

22. The conclusions do not provide any new insights. We know that Q and P are correlated and that the computation of joint probabilities is associated with uncertainties. I think that this study needs to be redesigned in order to provide some insightful contribution.

23. The Figures need some attention. Figure 1: would profit from focusing on the most important information, I would use a white-gray gradient for the topography and color all the study catchments in the same color. Or do the colors have a meaning? Figure 2: are the points shown the catchment outlets or centroids? I would use one color gradient for both mean AM and mean Q to make them comparable. I would use one color with a gradient instead of rainbow colors because one increasing variable is shown. For more detailed advice on how to design maps, please refer e.g. to Designing Better Maps: A guide for GIS users by Cynthia Brewer. This comment refers to all the following maps as well, which also just display one variable and where a color gradient of one color would be the more appropriate choice. Subpanels should be labeled with (1), (2), … or (a), (b),… to facilitate referencing.

**Minor points**

- L. 46-47: are you trying to talk about bivariate frequency analysis here?
- L. 49: Indicator approaches?

- L. 57-58: flood peak, duration and volume are different characteristics of the same event. I would not call these compound.
- L. 85-90: The paper follows a pretty standard structure of Introduction, Methods, Results, Conclusions and this overview is therefore not needed.
- L. 115: what is meant by automatic variogram choice?
- Description of Figure 6: Are magnitudes displayed here for the three return periods 10-, 50- and 100 years? This is not entirely clear.
- L.251-256: belongs to the methods section.
- I do not see the purpose of subfigure 9b.

**References used in this review**

Brunner, M. I., J. Seibert, and A.-C. Favre (2016), Bivariate return periods and their importance for flood peak and volume estimation, *Wire's Water*, *3*, 819–833, doi:10.1002/wat2.1173.

Gräler, B., A. Petroselli, S. Grimaldi, B. De Baets, and N. Verhoest (2016), An update on multivariate return periods in hydrology, *IAHS-AISH Proc. Reports*, *373*(2013), 175–178, doi:10.5194/piahs-373-175-2016.

Kendall, M. G. (1937), A new measure of rank correlation, *Biometrika*, *30*(1/2), 81, doi:10.2307/2332226.

Poulin, A., D. Huard, A.-C. Favre, and S. Pugin (2007), Importance of tail dependence in bivariate frequency analysis, *J. Hydrol. Eng.*, *12*(4), doi:10.1061/(ASCE)1084-0699(2007)12:4(394).

Serinaldi, F., A. Bardossy, and C. G. Kilsby (2015), Upper tail dependence in rainfall extremes: would we know it if we saw it?, *Stoch. Environ. Res. Risk Assess.*, *29*, 1211–1233.

---

## Referee Comment (RC2) · Anonymous Referee #2 · 8 Mar 2020

The manuscript analyses the join return period of extremes in precipitation and extremes in specific discharge. It is a point of view if considering precipitation and discharge as compound events or as a causal relationship. After Granger (1969), causality is given if one event is triggering another and the effect is occurring after the cause event. Nevertheless of recent discussions of this concept and the increase of data science methods for inferring relationships in cases where only few theoretical knowledge is available for defining cause-effect relationships, discharge can still be defined as an effect of precipitation. Moreover, there is sound theoretical knowledge on this relationship available and rainfall (cause) is in nearly all forecasting systems used as a causal variable for predicting discharge. From this perspective, I don't see any added

value of the proposed method. The authors do not state why they have chosen the presented approach. Second, the analysis was done on gridded data. Specific discharge was interpolated spatially across catchment boundaries. Mostly, specific discharge is a catchment-related characteristics and the spatial interpolation of this value should consider only areas upstream of the measured values and should not cross catchment boundaries. The analysis of the pairs "precipitation-discharge" at grid-cell basis does not account for upstream processes. This methodological setup would be very interesting for analyzing compound pluvial floods (local rainfall extremes) and fluvial floods (discharge resulting from cumulative upstream rainfall) but it is in my opinion not suited for answering the stated research question. In summary, I do not recommend the publication of the manuscript in its present setup. Thus, I don't go into further details.

Granger, C.W.J. 1988, Some recent development in a concept of causality. Journal of econometrics 39.1-2: 199-211.

---

## Author Comment (AC1) · 11 Mar 2020

The authors want to thank both Reviewers for the critical comments and suggestions. After a long discussion with co-authors we decided that a proper respond as well as a necessary revision of the manuscript's content and structure will most probably get well beyond a given timeline. For that we expect almost a complete change of the set-up concept and methodology. Therefore, we ask the Editor to withdraw the submitted pre-print.

---

## Editor Comment (EC1) · Nadav Peleg (Editor) · 12 Mar 2020

Dear Authors,

I received your request to withdraw the manuscript. I have confidence that the comments and suggestions of the two anonymous reviewers that are posted in the open forum (and a third report by Francesco Serinaldi, which is no longer available in open discussion) will assist you in improving the manuscript.

Sincerely,

Nadav Peleg